# Naturalising Agent Causation

**DOI:** 10.3390/e24040472

**Published:** 2022-03-28

**Authors:** Henry D. Potter, Kevin J. Mitchell

**Affiliations:** 1Smurfit Institute of Genetics, Trinity College Dublin, D02 VF25 Dublin, Ireland; hpotter@tcd.ie; 2Institute of Neuroscience, Trinity College Dublin, D02 PN40 Dublin, Ireland

**Keywords:** agency, autonomy, holism, reductionism, historicity, indeterminacy, multiple realisability, meaning, normativity, event-causation

## Abstract

The idea of agent causation—that a system such as a living organism can be *a cause of* things in the world—is often seen as mysterious and deemed to be at odds with the physicalist thesis that is now commonly embraced in science and philosophy. Instead, the causal power of organisms is attributed to mechanistic components *within* the system or derived from the causal activity at the lowest level of physical description. In either case, the ‘agent’ itself (i.e., the system as a whole) is left out of the picture entirely, and agent causation is explained away. We argue that this is not the right way to think about causation in biology or in systems more generally. We present a framework of eight criteria that we argue, collectively, describe a system that overcomes the challenges concerning agent causality in an entirely naturalistic and non-mysterious way. They are: (1) thermodynamic autonomy, (2) persistence, (3) endogenous activity, (4) holistic integration, (5) low-level indeterminacy, (6) multiple realisability, (7) historicity, (8) agent-level normativity. Each criterion is taken to be dimensional rather than categorical, and thus we conclude with a short discussion on how researchers working on quantifying agency may use this multidimensional framework to situate and guide their research.

## 1. Introduction

When an organism acts in the world, is it right to say that *the organism* caused the effect? Or is that simply a useful metaphor or a convenient level of description? Perhaps the more accurate statement is that some biochemical pathway or neural activity *within* the organism caused the effect. In other words, is it right to think of organisms as agents capable of action? Or are they simply loci of complicated happenings that give the illusion of concerted, autonomous agency? Are there things that happen in the world that are rightly said to be “up to” organisms, things that *they do* as agents, or are such happenings, in fact, reducible to the physical evolution of the components that constitute the organism? These are the questions posed by the concept of agent causality.

Introduced by Thomas Reid in 1863, “agent causation” is the claim that agents themselves can be the cause of events [1]. That is, organisms or systems *as a whole* can have causal power that is not entirely reducible to the causal power of their component parts or determined by the states of the environment [2,3,4,5]. Therefore, it mirrors a much older concept, framed by Aristotle and Epicurus in the context of human beings, of some things being “up to us”.

To justify agent causation, then, systems need to exhibit a causal power that *irreducibly* inheres at the level of the whole system (i.e., it is nonreductive) while still maintaining that the causal power is instantiated in, or realised by, the system’s physical constituents (i.e., it obeys physicalism). Timothy O’Connor describes this as an “ontologically primitive causal power” that is “at once causally dependent on microphysically-based structural states and yet ontologically primitive” [5] (p. 195)—where ‘primitive’ refers to this notion of irreducibility, rather than its meaning in evolutionary terms. For many philosophers, this is logically incoherent, and so agent causation is deemed to be conceptually and metaphysically impossible or at least highly problematic [6,7,8]. To see why, one needs to consider a number of different but overlapping ontological and methodological arguments that each appears to refute the concept from slightly different angles.

The first is the idea of *vertical reductionism* and the associated *causal fundamentalism* that typically comes with it. Vertical reductionism is the idea that for every macroscale description of a phenomenon, there is a microscale description that fixes it. In philosophy, this is often referred to as a supervenience relation: the macroscale supervenes on the microscale if a change in the microscale is necessary for a change in the macroscale to occur. Consider, for example, the relation between an individual molecule and the atoms that underlie or fix it. The properties of the molecule do not change without change at the atomic scale. If the microscale fixes the macroscale, then there is a sense in which the macroscale can always be derived from, or explained in terms of, the microscale. That is, any larger scale description of a phenomenon can be reduced to a description that uses smaller scales.

The logical endpoint of this vertical reductionism is that everything is describable in terms of the smallest possible physical elements in the universe—quantum fields, perhaps—and the interactions between them. This poses a threat to agent causation because the natural instinct is to infer that, because everything at a larger scale of description, including agents, is derivable from the properties of and interactions between quantum fields, what causal work is there left to do at the macroscale? Or as philosopher Jaegwon Kim puts it:

“If an emergent, *M*, emerges from basal condition *P*, why cannot *P* displace *M* as a cause of any putative effect of *M*? Why cannot *P* do all the work in explaining why any alleged effect of *M* occurred?”[7] (p. 558).

If all causation is fixed by microphysical happenings, then agents must be epiphenomenal. There is simply no room in the universe for macroscale phenomena to have the kind of ‘irreducible’ causal power needed for agent causation [6,9,10].

The second challenge to agent causation comes from what we could call *horizontal reductionism*. Here, the threat to agent causation stems from claims that it is not the system, as a whole, that has causal power. Instead, causation is attributed to a specific subset of components *within* the system—it is a particular ‘part’ (or set of parts) that causally determines the action of the ‘whole’. Thus, the whole does not have any causal power in virtue of being a whole because causation is entirely localised to (a set of) components with smaller spatial and temporal dimensions than the whole.

The important difference between vertical reductionism and horizontal reductionism, then, is that, in the former, the macro- vs. micro-distinction (*what* is being reduced to *what*) is between scales of description. However, crucially, at whatever scale you are using, the description is always of the *entire* system, that is, pitched at the spatiotemporal dimensions of the whole. Whereas, in horizontal reductionism, the macro- vs. micro- distinction refers to what is being described. The important causal elements in the system are reduced to a subsystem, or set of parts, that is spatiotemporally *smaller* than the system itself. Importantly, this can be carried out using any description scale. For instance, if the system in question is a brain, you could describe it in terms of all the atoms that constitute the brain and then causally reduce that to a localised subset of those atoms, or you could describe the brain in terms of its neuronal connectome and then causally reduce that to a localised subset of neural states.

The traditional alternative to agent causation, *event causation*, is one form of horizontal reductionism. It claims that an organism’s actions are caused by states and events that simply *involve* the organism [11,12]. Event-causalists tend to cite a particular psychological state (e.g., a belief) as the sufficient cause of a given action, with these states, themselves, having previously been caused by antecedent states or events. By horizontally reducing the locus of causality to events and states *inside* the organism, this view leaves the ‘agent’ out of the picture entirely. Agents are not identical with a particular belief or desire or goal, so if these psychological states wholly determine the system’s behaviour, then there is no sense in which the ‘agent’ can be said to have caused its effects in the world. This is sometimes referred to as the ‘problem of the disappearing agent’ [3,13,14].

*Eliminative materialism* goes one step further than event causation, claiming that certain psychological states such as beliefs and desires do not exist and, thus, we should only appeal to neuroscientific explanations when identifying the cause of an organism’s action [15,16]. This position is, therefore, both vertically *and* horizontally reductive; in that it (vertically) reduces psychological states to neural states and (horizontally) reduces the whole system to a subset of neural states within the system. One issue worth noting is that it is not clear why eliminative materialism does not lead to the causal fundamentalism that typically comes with embracing vertical reductionism. In other words, what makes neural states the right scale of description at which to identify causation? If a reduction from psychological states to neural states is permissible, why not reduce further to a causal explanation at the scale of cellular components, electrical ions, or the subatomic particles and quantum fields that compose those neural states? As we have seen, once the door to vertical (causal) reductionism is opened, it is hard to see how one could resist the fundamentalist pull to only identify causation at the smallest scale of physical description.

The third and final threat to agent causation is from *external determinism*. If systems are determinately ‘pushed around’ by states and events in the environment, then to what extent can the system be meaningfully considered the cause of its actions? Instead, agents need to be causally autonomous from their environment, in that they have “the ability to do what one does independently, without being forced to do so by some outside power” [17] (p. 305). In this sense, external determinism parallels, and, in fact, overlaps with, the horizontal reductionist challenge to agent causation, since both threaten to ‘explain away’ the agent by localising causal power to somewhere that is *not* the system as a whole. In horizontal reductionism, it is events and states within the system that wholly determine the system’s actions. In external determinism, it is events and states in the environment that wholly determine the system’s actions.

These three lines of argument, vertical reductionism, horizontal reductionism, and external determinism, appear to underlie the general consensus among philosophers that agent causation is not tenable. Our primary goal in this paper is to demystify and revive the concept of agent causation by presenting a set of conditions that, in principle, would enable a theoretical system to overcome all three of those arguments if met. In other words, our aim is to propose a set of general criteria that may collectively justify ascribing agent causation to a system of study. These are:Thermodynamic autonomy;Persistence;Endogenous activity;Holistic integration;Low-Level indeterminacy;Multiple realisability;Historicity;Agent-Level normativity.

Our intention is not necessarily to argue that these criteria are complete and definitive but rather to demonstrate a plausible way in which agent causality can be conceptualised and realised in systems without violating the thesis of physicalism. In doing so, the secondary aim of this paper is to investigate and argue that living systems satisfy at least some of these conditions. Note that we do not consider the criteria as bright lines but rather as dimensions along which different organisms may vary. Moreover, though we highlight each condition separately for the purposes of this paper, they do, in fact, overlap and co-depend to quite a considerable degree. While each is a necessary condition in its own right, the full sense in which the eight criteria may justify agent causation comes from understanding how they fit together as a collective.

We should also note that we take agent causation to be necessary but not sufficient for ‘free will’. The criteria presented here are intended to provide a naturalised account of how a system may exhibit agent causation, but we do not assume that such a system would necessarily have free will (where the former is loosely understood as ‘doing what you want’ or ‘acting for your own reasons’, while the latter also includes the meta-capacity to ‘want what you want’ or ‘reason about your own reasons’ [18]).

In short, we hope to (i) convince readers that agent causality is a plausible and appropriate way to think about causation in biological systems, and (ii) set out a conceptual framework for a more productive and empirically grounded investigation into the concept of agency within biology.

## 2. Criteria for Agent Causation

### 2.1. Thermodynamic Autonomy

One presumably uncontroversial condition for agent causality is that the system in question is thermodynamically distinct from its environment (known, technically, as being *out of equilibrium* with the environment). Indeed, this is necessary to be an organism or an entity at all—let alone be an agent or a *cause* of events in the world. Systems, first, have to be an identifiable thing, with a physical boundary that legitimately separates them from the rest of the universe [19,20,21]. As Keith Farnsworth explains:
“If there were no physical boundary between an agent and its surroundings, then any external force would act unaltered throughout the agent, and any force initiated by the agent would act equally on the internal and external environment, so no distinction in action could be made between the agent and its surroundings. If no distinction can be made, then no measurement could be made to tell us whether an action arose from within or beyond the agent”[22] (p. 12).


Living systems achieve this separation from the environment via a cell membrane or an outer skin in more complex organisms. These physical barriers define the system as an entity, with a discernible inside and an outside, and, at the same time, enable the system to do work to remain out of thermodynamic equilibrium with the environment. The physical barrier also grants the organism a degree of causal insulation from the external milieu. Instead of being exposed to every flow of physical and chemical causation that comes its way, the system is sheltered from that storm and able to buffer its internal dynamics. This is a crucial first step toward agent causation because it means that events in the environment are not determinately pushing the system around. External forces may impinge on the system without leading to a change in its behavior; thus, creating space for other (agent-level) factors to have a causal influence. In this sense, a physical barrier lays the groundwork for systems to overcome the threat posed by external determinism (see Section 2.3).

Moreover, the system’s barrier opens up the possibility of a new form of causation, dependent on information. Rather than letting (potentially noxious) chemicals from the environment into the system, specialised receptor molecules can sit in (e.g.,) a cell’s membrane. These receptors have both external and internal components, such that when the external end binds a chemical, it causes the entire molecule to change its conformation, which, in turn, can impact the dynamics *inside* the cell. No matter or energy passes through the barrier. Rather, the receptor transmits a *signal* that carries information about something out in the environment, namely, the presence of chemical X. The information is what causally impacts the system’s internal processes and—as we explore below—this new form of *informational causation* can be heavily influenced by the current state of the cell (see Section 2.3 and Section 2.4) and its historicity (see Section 2.7); such that the behaviour of the system is not wholly explainable in terms of environmental causes alone (*contra* external determinism), nor wholly understandable in terms of sublocalisable, instantaneous physical states (*contra* horizontal and vertical reductionism). Thus, the requirement for agents to be thermodynamically autonomous systems sets the foundation upon which most, if not all, of the other agent causation criteria, are built.

### 2.2. Persistence

Another straightforward and foundational requirement for agenthood is that the system in question persists through time. While the duration necessary to satisfy this condition is an open (and probably unproductive) question, at least some degree of persistence is needed in order to be around long enough to cause an effect. Moreover, as we will discuss in Section 2.7 and Section 2.8, persistence on phylogenetic and ontogenetic timescales are both required for systems to exhibit the type of *temporally extended* causation that, we argue, is needed to overcome the threat of eliminative materialism.

Living systems clearly satisfy the condition of persistence. Indeed, life is often characterised as the ability to (locally) resist the second law of thermodynamics and survive for extended periods of time [19]. Unlike rocks which persist passively, due simply to physical hardness and chemical inertness (i.e., through stasis and inertia), biological persistence is dynamic in nature. Organisms obtain stability out of a constant internal flux of recursive chemical reactions and self-sustaining causal loops, where component parts do work to constrain one another within the bounds necessary to keep the whole system going. Living organisms thus persist as dynamic, holistic patterns that constantly regenerate the constraints required to keep themselves organised [23,24,25,26]

There are two important consequences of the organism’s method of persistence. First, the system needs the energy to fuel the self-sustaining cycles of constraints. Organisms must therefore remain open to the environment in a controlled way, locating food and converting it into energy that can be used to maintain the ongoing internal dynamics. Second, life is a pattern. What persists through time is the organisational pattern of self-maintaining, dynamical processes, *not* the physical material that realises that pattern at any one moment. In fact, the atoms that constitute the system are regularly being replaced, yet the dynamical pattern persists, and we take this to be the organism persisting.

This property immediately undermines horizontal reductionism since it is not clear how one could isolate causation to a subset of specific elements *within* the pattern when holistic integration is the system’s defining characteristic (see Section 2.4).

### 2.3. Endogenous Activity

Non-equilibrium thermodynamic systems that persist through time can stand apart from the physical world long enough to be a causal factor in it. However, to justify agent causality and meaningfully be labelled the *cause* of their effects, systems also need to sidestep the challenge of external determinism. Their actions cannot simply be determined by events in the environment, such that one could predict the behaviour of the system from external states alone. In other words, agents need to have some degree of causal autonomy from the outside world.

As we saw in Section 2.1, a physical barrier is a good start: it insulates the system from external perturbations so that not every change in the environment causes a change in the system. However, it does not follow from this that every change in the system is not caused by a change in the environment. Thus, a physical barrier is not sufficient for causal autonomy since it is still the case that the forces that *do* causally impact the behaviour of the system could do so in a deterministic, linear fashion (as suggested by external determinism). 

We propose that a condition for agent causality is not just persistence but *active* persistence. Systems need to internally initiate their actions, to some degree, so as to avoid being mere stimulus-response machines that are passively pushed around by the environment. In this way, external factors can *influence* but not determinately cause the system’s behaviour in a manner that would be incompatible with agent-level causation (due to external determinism).

As we outlined in Section 2.2, a defining feature of living organisms is their dynamic, internal activity. They are constantly doing thermodynamic work internally to self-organise and maintain the pattern of processes that define their existence. To this end, organisms are better understood as actively monitoring and *adjusting* to information about conditions in the environment rather than being pushed around by these conditions. External inputs are assimilated into ongoing patterns of biochemical and/or neural activity rather than driving that activity itself.

Evidence of this sort of endogenous activity comes from all areas of biology. Central pattern generators [27], for example, which *spontaneously* generate rhythmic oscillatory patterns in the brain, are found in almost all vertebrates and invertebrates and play an important causal role in organism-level behaviours such as walking, swimming, and flying [28]. In humans, our internal activity in the absence of external stimuli is abundantly clear whenever we introspect or sit alone with our thoughts. This has been shown experimentally through reports of visual hallucination in sensory-deprivation contexts [29,30,31]. The brain’s constant activity is also evident in brain-imaging studies: fMRI research showed that task-specific neural activity typically amounts to just 1–2% of background brain activity in relevant areas and, thus, energy-consumption at rest is almost the same as during a demanding task [32] (see [33] for a full overview of the brain as an endogenously active organ). 

Taken together, this evidence presents a clear picture of organisms as endogenously active systems. Even when they appear from the outside to be at rest, they are not static internally [34]. They are constantly doing work to readjust their internal configuration so as to keep the whole dynamic, self-maintaining process alive and persisting. This equips the system with a degree of causal autonomy from the environment: external stimuli can influence the system’s behaviour, but only within the context of its current internal dynamics. That is, the system’s actions in the world are ultimately generated from *within*, with information from the environment used to guide these dynamics as needed [35].

Another point to note here is that the information that organisms receive from their environments is very rarely sufficient for them to unambiguously determine an appropriate course of action. Often, external inputs are simply not rich enough for external determinism to hold. Instead, organisms actively probe the environment, gathering information, making inferences about what is out in the world, and trying out new problem-solving techniques. Thereby providing clear instances of endogenous action that is not determinately driven from the outside [36]. In this sense, even perception might be understood as a form of internally initiated action [37].

Therefore, living systems provide a model for conceptualising how systems, in general, can overcome the challenge posed by external determinism and, thus, move us one step closer to agent causality. In the rest of this paper, we will consider how a system might overcome the reductionist threat, starting with horizontal reductionism in the next section.

### 2.4. Holistic Integration

The horizontal reductionist challenge to agent causation paints a picture of systems as machine-like, made up of isolatable mechanisms, each of which performs its own specialised function, which are then combined in a serial, linear, additive fashion to create the system (or *cause* it to come into being). This sort of explanation is “analogous to a recipe for producing a phenomenon starting from a list of ingredients, where the ingredients are mechanistic entities and their properties, and the recipe amounts to the organization and sequence of activities these entities perform” [38] (p. 105). This perspective leads to the ‘agent’ being left out of any causal explanation for its own behaviour because, for any given action, there is an identifiable mechanism, or linear cause-effect pathway, *within* the system that can be pointed to as the cause of that action. Thus, while the system may have escaped being deterministically pushed around by external events, it is now being deterministically pushed around by some of its own component parts instead. There is no room left for a causal power that inheres at the level of the whole system (i.e., agent-level causation).

However, we contend that not *all* systems can be coherently decomposed into separable parts in this machine-like manner. As we saw above, living organisms are dynamic, holistically integrated systems whose parts constantly act in concert, influencing and constraining one another in order to maintain the holistic pattern. Even the simplest organisms show substantial degrees of integration. For example, bacterial chemotaxis, when a bacterium locomotes up or down a chemical gradient in its environment, is one of the simplest and most well-studied behaviours in biology. There exists a well-understood pathway in the bacterium that links a transmembrane receptor for detecting food substances, via an internal signal transduction cascade, to a flagellum that controls its motion [39]. The temptation is thus to explain bacterial chemotaxis in terms of a simple linear chain of fully determining causes and effects, starting with the detection of a chemical stimulus and then sequentially moving through each part of the chemotactic pathway until a locomotion behaviour is performed. However, the apparent discreteness and linearity of this pathway, as suggested by highly controlled (and thus, highly artificial) experiments, is somewhat illusory. In fact, contextual information about the current metabolic state of the cell constantly integrates with and modulates the chemotactic pathway [40] in a way that is highly nonlinear and that challenges any suggestion that the pathway determinately causes chemotaxis. If changes in the metabolic state of the cell can change how the action is executed, despite the same individual stimulus being present, then it does not seem accurate to isolate the chemotactic pathway and identify it as the cause of the action. Such an interpretation only arises when one holds the metabolic state of the cell constant while studying how bacterial chemotaxis works. As we have seen, this would afford only a partial understanding of the mechanism because it misses the system’s holistically integrated and relational structure, as well as the endogenous activity of the system.

Bacteria can also integrate a number of different environmental cues simultaneously, including pH levels, temperature, and osmolarity, and use information from the past to inform and influence action [41]. It was even shown in *E. coli* that crosstalk between one signaling pathway (heat shock) could modulate the activity of a significantly different pathway (respiration) [42,43]. Moreover, the direction of bacterial chemotaxis itself relies on temporal integration. The bacterium does not act based on absolute levels, but according to information it gathers about concentration changes as it moves. The resultant picture is one in which, even in this very simple organism, “sensory, regulatory, and metabolic networks must all drive environmental perception and corresponding action” in a strikingly holistic and integrated fashion [44] (p. 364). The argument that the bacterium’s behaviour is determined by any identifiable ‘part’ or even an isolatable pathway is thus difficult to maintain when the whole organism is clearly involved in the regulation and execution of this behaviour.

We see increasing degrees of holistic design as we scale up to more complex organisms with brains and nervous systems. Fuelled by recent technological advances that allow researchers to record neural activity on a much larger scale than ever before, we are now learning that brain areas we once understood as single-mindedly carrying out their work in relative isolation are, in fact, highly sensitive to the activity in other brain areas. Visual cortex activity, for example, as well as processing visual information, is substantially modulated by an organism’s movements; with evidence coming from running and hindlimb flexions, all the way down to small orofacial movements and pupil dilations [45,46,47]. Indeed, information about actions, goals, diverse sensory percepts, internal states, and other parameters is much more widely distributed than previously thought, meaning that all local signaling is modulated by the context of global brain states [48,49].

Similarly, the brain regions, circuits, and processes that mediate decision-making and action selection are highly distributed, involving ongoing signaling between multiple subsystems across the brain, working in parallel in recursive, interlocking cycles over some duration of time [48,50,51,52]. The result is a dynamic, system-wide interaction, in which parts are continuously modulating and constraining each other until they all *collectively* settle into a new state. To decompose this process into a set of functionally independent (machine-like) ‘parts’ carrying out their work in isolation, and then combining to cause the system’s next state, is to entirely miss the essentially dynamic and holistic nature of it.

Therefore, the evidence suggests that biological systems are too holistic, too integrated, and too relational to submit to a machine-like analysis. Instead, they are more akin to Stuart Kauffman’s concept of a ‘Kantian Whole’ (which, as the name suggests, derives from the work of Immanuel Kant in his *Critique of Judgement*). Kauffman depicts a system in which “the parts exist for and by means of the whole, and the whole exists for and by means of the parts” [53] (p. 609). That is, a system so deeply interconnected that it does not make sense to decompose it into its component parts because the true essence of the system exists in the relations between those parts [54]. In this sense, organisms offer us a way to conceptualise a system that does not fall prey to the horizontal reductionist challenge. If a system is so holistically integrated that to understand any given part, you must also understand the whole, then causal power will meaningfully inhere at the level of the whole. You cannot horizontally reduce such a system to identify a particular part (or set of parts) that is determining the system’s next state because the activity of that part is, itself, determined by all the other parts in the whole. Therefore, we suggest that holistic integration is a necessary condition for agent causality.

In the final section of this paper, we consider where the instinct towards horizontal reductive thinking comes from in the practice of science and philosophy. Below, we explore the properties of organisms that argue against vertical reductionism.

### 2.5. Low-Level Indeterminacy

If the arguments above deflate *horizontal* reductionist claims, it could still perhaps be argued that the real causation inheres at the lowest level, in the physical interactions of the components, even if they have to be considered as a single, unified dynamical system (no matter how complex those interactions might be). Under this *vertically* reductionist view, the state of the entire physical system at time *t* (which may need to include the state of its environment), plus the low-level laws of physics, fully determine the state of the system at some subsequent time, *t* + 1. Logically, this would extend to *t* + 2, *t* + 3… *t* + *n*, and on to infinity. This Laplacian view (of complete determinism from the dawn to the end of time) is encompassed in Kim’s argument that ‘every physical effect has a sufficient physical cause’, with the implication that those causes are necessarily located at the lowest levels (or smallest scales of description) [6,55]. This argument thus asserts both physical pre-determinism and causal reductionism. If the laws determining the interactions of particles or the evolution of quantum fields are causally comprehensive, then no higher-order causes can be admitted.

However, the evolution of quantum fields is not fully predetermined in this way. Though the Schrödinger equation gives a definite solution for how quantum fields comprising any system will evolve, this solution is a distribution *of probabilities* [56]. As soon as some interaction occurs of the type necessary to actually observe the state of the system (but not actually relying on an observer, per se), some particular set of these probabilities will be realised. What determines what set becomes realised (within the probability parameters as defined) seems to be truly random—unpredictable *in principle*, not just in practice, consistent with the Heisenberg Uncertainty (or Indeterminacy) Principle. Though there are very different interpretations of what this means for the underlying nature of reality [56], the upshot is that the totally defined microscopic state of a complete system, together with the fundamental equations that comprehensively describe the evolution of quantum fields, as an empirical fact, *do not* deterministically predict the next state of the system [57,58]. This thereby brings into doubt the causal fundamentalist urge to identify all causal power at the system’s lowest level of description. 

This fundamental indeterminacy is, therefore, a necessary but not sufficient condition for the emergence of some kind of higher-order causation [4,59,60]. Many physical systems will evolve simply according to the partly random realisation of these underlying probabilities. However, others may come to be structured in such a way that constrains their evolution according to some kind of higher-order, functional criteria. This is exactly what happens in living organisms, due, in the first instance, to the iterative, ratchet-like action of natural selection and, second, to the ongoing learning that individual organisms engage in over their lifetimes. Both of these processes select for structures that physically embody criteria for action (see Section 2.6), enabling organisms to accumulate causal power and act in ways that promote their own persistence (see Section 2.7).

This goal-directedness of living organisms means that higher-order states can have meaning and value (normativity) for the organism (see (Section 2.8). The system can be configured in such a way that it represents and operates on information—physical states that are *about something* and which inform the actions of the agent. This kind of informational causation thus enables agents to do things for reasons. However, if this view is to be defended, the causation must depend on the meaning of higher-order patterns, which, though it must be realised in some low-level state at any moment, cannot be reduced to low-level details. Here, the concept of multiple realisability is crucial.

### 2.6. Multiple Realisability

A key feature of agency is doing things *for reasons*. The lowest-level details of a system, e.g., the neurons in a brain or the atoms and molecules that constitute them, do not have reasons; certainly not *reasons for* system-level action, at least. Therefore, a system cannot justify agent causality if it is being entirely driven around by the constituents at its lowest level of description. Instead, higher levels need to have some degree of causal autonomy from the lowest-level parts, such that changes at the lowest level do not *necessarily* cause changes at the higher level in a simple feedforward or bottom-up fashion. This is a crucial step toward agent causality because it means that at least some of the system’s causal power must inhere at the higher levels of organisation, where it might start to make sense to talk about agent-level *reasons*.

The structuring of criteria in living organisms, which determine the flow of information and interpretation of different patterns of activity, embodies exactly this kind of higher-order causation [61]. This is particularly elaborated in neural signaling, where communication from one neuron to another, or one population of neurons to another, depends on distinct macroscopic patterns, which each can be instantiated or realised in many different microscopic arrangements (i.e., multiple realisability).

Take the communication between neurons, for example. Neuron A receives inputs from Neuron B via neurotransmitter molecules that are released when B fires. These molecules can be bound by neurotransmitter receptors on Neuron A, which open, allowing sodium ions to rush into the post-synaptic structure. This generates a change in the electrical potential gradient between the inside and the outside of the cell. However, this charge is rapidly buffered unless the total increase over some short time period pushes the potential above a threshold, which triggers an explosive amplifying event—the action potential or firing of the neuron, transmitting an electrical signal to the other neurons it has synaptic connections with (also known as a spike). If the threshold is *not* reached, the neuron will not fire, and the electrical potential will peter out, in effect wiping any record of neuron B’s original spike.

Neuron communication is, therefore, generally not designed to be a faithful or exact transmission of information but rather an interpretation of it. Some spikes will be completely ignored and for those that are not, details such as the timing of a spike are still often ignored, i.e., the specific details of exactly *when* the spike occurred simply do not matter (outside of some specialised systems such as auditory neurons). Instead, neuron A is only interested in whether the number of spikes that come its way within a given time frame is sufficient to satisfy its threshold, or, as neuroscientist Peter Tse frames it, sufficient to meet the criteria neuron A places on its inputs [61]. The process is thus highly nonlinear; in particular, many changes in the input do not lead to changes in the output. Individual neurons are configured so that they monitor their inputs in different ways; some act as temporal filters, waiting for strong bursts of inputs over a limited time window before firing, others act as coincidence detectors, where near-synchronous spikes from multiple input neurons are needed to meet their conditions for action. The low-level details thus often do not matter, and they are often lost in this coarse-graining process. As a consequence, two patterns of input spikes with different arrangements may effectively prompt the same activity in the downstream neuron. In effect, they mean the same thing to that neuron, and it is the meaning of the pattern that has causal efficacy.

The same principle applies at the level of populations of neurons [62,63]. Because of the network of excitatory and inhibitory connections among any local population of neurons, the possible patterns of activity that groups of neurons can exhibit is constrained [63,64,65]. Neuronal populations within a brain region consequently tend to occupy a particular set of patterns, or ‘attractor states’, far more often than any of the other patterns they could conceivably occupy. This means that the potentially high-dimensional state space that the population could occupy is constrained to a much lower-dimensional manifold. Brains are shown to capitalise on the occurrence of these population-level attractor states to, once again, use higher-order information to drive the system, rather than rely on the low-level details (in this case, the states of the individual neurons). A given population can exhibit a number of neural modes or subspaces of the possible multidimensional activity space, only a subset of which may be efficacious in driving activity in downstream areas [66]. Perhaps a more helpful way to frame this relationship is that the second population of neurons is configured to monitor activity in the first region and selectively activate when certain patterns appear while effectively ignoring others [67]. A different downstream population may only activate when it detects a different set of patterns, depending on what information it is “interested in”.

These functional criteria can be prewired by evolution or crafted through learning. In either case, they are aligned to the goals of the organism, better enabling the organism to do the causal work required for its own persistence. Moreover, they can also be modulated on the fly by processes of attention or top-down selection, realised through rapid synaptic reweighting [61]. The functional criteria that direct each neuron’s or each population’s interpretation of any given inputs are thus not fixed or isolatable but depend holistically on global context (thus, also *contra* horizontal reductionism).

Multiple realisability of this sort undermines claims that the causality really rests in the details of the lower level. The vertically reductive argument is that even if some low-level pattern becomes coarse-grained in transfer to the next neuron or population, since the macrostate must, at any moment, or over some defined duration, be instantiated in *some* particular microstate or sequence of microstates (i.e., it supervenes on the microstates), then any particular microstate will necessarily fix the macrostate. Therefore, the causality that we had attributed to the macrostate A (under the relationship: if A, then X) could really be completely inherent in the corresponding microstate, A′, which would also entail X. However, there is a reciprocal relationship that is usually implied in this kind of causal relationship. If A is supposed to be *the cause of* X, then it is usually understood that the counterfactual NOT A would imply NOT X.

Under multiple realisability, that counterfactual applies to the macrostate but not necessarily to every microstate. If the microstate is changed from A′ to B′, corresponding to macrostate B, rather than A, then you may obtain a different outcome, Y. However, if the microstate is changed from A′ to A′′ or A′′′, all of which still corresponds to macrostate A, then you would *not* see a change in the outcome; X would still occur. The effect, X, is therefore ‘sensitive’ to the multiply-realisable macrostate, A, rather than the microstate that physically realises it (i.e., A′, A′′ or A′′′) [68]. The causal information, under this kind of counterfactual reasoning, is thus rightly said to inhere at the level of the macrostate, i.e., in the higher-order pattern, rather than the low-level details.

Taken together, low-level indeterminacy and multiple realisability appear to bypass the vertical (causal) reductionist challenge to agent causality. Low-level indeterminacy shows that the lowest level of description is not deterministically fixing the next state of *any* system, while multiple realisability represents a way in which *some* systems can capitalise on this indeterminacy in order to exert a form of higher-order causation. In the sections below, we argue that these multiply-realisable, higher-order patterns have causal efficacy in the brain by virtue of what they *mean* for the organism. This provides a framework for naturalising *reasons* in systems more generally.

### 2.7. Historicity

Above, we described how low-level indeterminacy creates some causal slack in the system that can enable the emergence of non-reductive (or higher-order) causation. But does this actually get us any closer to agent causation? Even though it is higher-order patterns of information that push the system around and not the lowest-level physical happenings, it is still not obvious how the causal power in the system could sensibly be understood in terms of *agent-level reasons*. Is the whole not still being ‘pushed around’ by its parts? It is just being carried out by patterns of neural activity now. In this case, there is still no sense that the ‘agent’ is causing its effects in the world.

To bring the agent into the picture, we need to reflect on our framework for understanding causation in contemporary science. Mostly, we limit our causal explanations to ‘how’ questions: we ask ‘*how* did X cause Y?’ Scientists then try to get to grips with the mechanisms that underwrite X’s behaviour, and we explain the causal relation between X and Y in those terms. However, an equally valid question to ask is: ‘*why* does X behave in that way?’ [69]. In the case of rocks or atoms or billiard balls, this is a somewhat uninteresting and unenlightening question, with the answer simply being the physical characteristics of the entity in question. However, in systems that run on information, it is a very fruitful line of questioning, precisely *because* these systems exhibit higher-order causation.

Consider, again, the neural population. An activation pattern among the neurons in this population is only causally efficacious (within the system) if there is a receiver or interpreter monitoring the population and activating in some way when the pattern appears [67]. That is, the pattern is only informational if there is something that is sufficiently configured to notice it. Therefore, it is relevant to ask *why* the interpreter is set up to detect the particular pattern it does (and subsequently activate in the way that it does) if we are to obtain a full causal understanding of the system. The answer, we suggest, is because of what the pattern *means*, not in any kind of immaterial or mystical sense, but in a way that is grounded in the system’s interactions with the world, shaped by its history, and instantiated in its physical structures.

In biology, natural selection infuses organisms with the purpose to persist. If organism A is set up such that it out-persists organism B, then future populations are going to look more like A than B. Over time, the set of surviving organisms is going to be the one that is best set up to persist; that is, the set of organisms whose value-system, what is seen as ‘good’ and what is seen as ‘bad’, is most optimally conducive to each organism’s persistence, given its historical environment (compared with all the other possible ways in which that value-system could be set up). In other words, a lineage’s successful persistence over evolutionary timescales cashes out as *meaning* or normativity in the individual. For example, the food chemical *means* something good to the bacterium because it was adaptive in the past to move toward it, and so it is set up to do so again. The noxious chemical *means* something bad because it similarly was adaptive to locomote away from it in the past, and so it is disposed to do so. This meaning is grounded in its ancestors’ previous interactions with the world and defined relative to their persistence; such that what the stimulus means to an organism is often well-aligned with the effect that stimulus is going to have on its chances of survival (based on the *actual* effect it had on previous generations).

In simple systems, the meaning of a signal can be pragmatically embodied through direct coupling to some historically adaptive action. The first nervous systems, presumably similar to the simple nerve nets observed in extant creatures such as Hydra and jellyfish, may have co-evolved with muscles to allow coordinated movement of the various parts of multicellular creatures. Linking these to sensory receptors could then allow appropriate responses to be selected based on information about the external world in the context of the animal’s own movements [70,71,72]. As organisms increased in complexity, perception and action became decoupled, with the addition of intervening layers now operating on internalised ‘representations’. These internal, representational patterns are still grounded by links to the periphery in both directions (from perception and to action); that is, they stand in exploitable relation to things in the world and adaptively inform action but they can now be integrated and operated on in more complex ways [73,74,75].

Returning to the neural population, we now see that the interpreter is set up to recognise and act in response to a given pattern *because* it is meaningful. If the pattern co-varies with a state of affairs in the world that can be used to enhance survival, then being configured to detect that pattern and use it to guide appropriate action in the world is adaptive. In many cases, the particular neural pattern that comes to represent or “be attached to” some referent is arbitrary, selected from a set of preconfigured endogenous neural ensembles that happen to be active at the time of some experience [35]. The semiotic relationships in the nervous system are thus not driven from the outside or determined by physical properties [76]. Yet, treating the pattern as meaningful and turning it into information that causally influences the next state of the system is going to lead to increased persistence, and thus, over time, lead to a higher percentage of the population being configured, not only to treat the pattern as meaningful, but for it to mean something relevant to action and survival.

In effect, for systems that run on informational causation and emerge from an evolution-like selection process that rewards persistence, it is *meaning* that drives the system and, thus, informs behaviour. Moreover, the meaning inheres not just in the higher-order pattern itself but also in the recognition and response to that pattern. This view thus undercuts vertically reductive, eliminative claims that behaviour can be reduced to the flow of patterns of neural activity, with the mental content or meaning of the encoded states being effectively epiphenomenal. In fact, it inverts that logic. Neural patterns have causal power in the system *solely by virtue of what they mean* to the organism as a whole.

Any attempt to understand or explain the causes of an organism’s behaviour is thus doomed to fail if it takes a purely instantaneous view of the physical system. It is not enough to account for how an organism behaves upon detecting some external stimulus or physiological state of affairs—the ‘triggering cause’. We must also understand why the system is configured such that it behaves in that way—the ‘structuring causes’ [69]. The actual causal influences are diachronic; that is, they extend through time. What gives organisms causal power is the evaluative record of their past experiences and those of all their ancestors (as well as, in a negative sense, all those unfortunate individuals who did not leave offspring). This causal power is hard-earned. Both natural selection and learning do causal design work, they configure the system in such a way that it embodies pragmatic knowledge about the world and itself that can be used to direct adaptive action. Living organisms thereby accrete causal power and come to act as causal agents in the world.

Crucially, agents do not just learn how to respond to things in the world and then sit waiting for those stimuli. They adapt their endogenously generated patterns of active behaviour to their environment and circumstances, learning what to learn from, what information to seek out, and what active tactics and strategies to use to best pursue their goals, crafting their own environments as they do so [77].

If we consider nervous systems as control systems, then we can broadly characterise the meaning of various neural states as representing beliefs (about things in the world or the state of the organism itself), desires (usually with short-term goals nested within a framework of longer-term goals), and intentions (possible actions that may be considered, evaluated, and ultimately selected for execution or not). Together with the foregoing discussion, this seems to yield a naturalised account of how a system can do something *for its own reasons*. Note that these reasons do not have to be conscious in order to justify agent causation (and mostly, they will not be). We contend that to be a system in which meaning is the causal driving force just *is* what it is to have reasons in the manner necessary for agenthood.

However, there is, a remaining challenge from some theories of event causation.

### 2.8. Agent-Level Normativity

In complex organisms with nervous systems, what determines the flow of brain activity from state to state is the *meaning* of the neural patterns. The idea that what an organism does can be reduced to the neural patterns themselves is therefore not tenable; the system is configured such that those patterns only have causal power by virtue of what they mean, i.e., the psychological states that they correspond to (conscious or subconscious). However, even if that inability to reduce decision-making to neural states is accepted, some proponents of event-causation argue that *psychological states*, beliefs, desires, and intentions, can do the necessary causal work. Under this view, the collection of such states at any given moment constitutes the ‘events’ that determine what happens next; the agent is simply the arena in which such states arise [12,78,79], (see [14] for an attempt to reduce the ‘agent’ to the event). 

We argue here that this view is incoherent. Beliefs, desires, and intentions are things that only an agent can have. Neurons do not have beliefs; neural circuits do not have desires; brains do not have intentions. Moreover, an intention is not a thing (either a substance or an event) that can either exist or have causal power by itself. It is the *agent having the intention* that has causal relevance to what happens. When the entire set of such psychological states is taken into account at any moment, holistically integrated, as described in Section 2.4, and freighted with historically grounded meaning as described in Section 2.7, we contend that this *just is agent causation*.

The agent itself is the locus of meaning. That is, meaning inheres at the level of the whole system, as an entity persisting through time, interacting with its environment and being judged on its behaviour. Percepts, drives, actions, and their consequences mean something for the whole agent, not for its parts. This derives from the fact that, from an evolutionary perspective, the whole organism is the locus of fitness, not its parts. Natural selection thus crafts the control systems of living organisms to detect, characterize, and operate on signals *as they are relevant for* the survival of the whole organism. The chemotactic system of *E. coli*, for example, is configured in the way that it is because it has been adaptive for millions of preceding generations to move up a concentration gradient of a food source. The value and meaning in that relationship are grounded relative to the purpose of the entire system, which is to persist as a unified whole.

In the transition from unicellular to multicellular life, the locus of fitness shifted from single cells to the multicellular whole. This transition involved a progressive division of labour within clonal organisms, first in transiently aggregating colonies and then in obligate multicellular creatures. In particular, the division of soma and germline means that individual somatic cells give up any chance of reproducing directly. However, their genetic material can be reproduced indirectly because it is shared with the germline cells. Natural selection thus ceases to care about the fate of individual cells in the multicellular organism; all that matters is that the organism as a whole survives and reproduces [80,81], and the functional roles of all cells are directed towards that end.

In animals with nervous systems, cumulative feedback from natural selection configures the nervous system to enable the organisms to detect stimuli and action possibilities in the environment that are most salient for the survival and reproduction of the whole organism. Perception is egocentric, action-oriented, and laden with value from the get-go. The goal is to create a map of objects out in the world that represent potential threats and opportunities *for the whole organism*. In parallel, the action selection systems are fundamentally configured around actions of the whole organism—approach, avoidance, exploitation, exploration—and whether those actions tend to be good or bad *for the whole organism*.

On both evolutionary and individual timescales, the agent is thus the locus of fitness. It thereby becomes both the locus of meaning and also the locus of control that is informed by and evaluated relative to that meaning. Of course, natural selection has not just given animals hard-wired instincts. It has endowed them with systems to learn from their experience, in particular systems of reinforcement learning that will up- or down-weight action choices based on the outcomes of prior behaviour. Again, the value of these outcomes, the thing that grounds the meaning of possible actions, is relative to and inheres at the level of the whole organism. Individual neurons do not feel rewards or punishments; the agent does, and it is the agent that is guided by these signals.

## 3. Summary

To summarise, we argued that the eight criteria outlined above constitute a completely naturalistic way for systems to, in theory, exhibit agent causation. To recap, systems can be agents if they are self-organising and causally insulated enough to persist through time, out of thermodynamic equilibrium with the environment. To avoid external determinism, they need to be intrinsically active, treating external inputs more as helpful information than determinate, causal forces. The proactive self-organising activity of these systems entails a holistically integrated structure, in which parts are too interconnected and context-dependent to be understood in a machine-like, decomposable, linear fashion (*contra* horizontal reductionism). On top of this, these systems can be driven by meaning and reasons because higher-order organisational patterns are able to coarse-grain over microphysical happenings by virtue of the existence of some degree of indeterminacy at lower levels. This meaning derives from temporally extended causal processes that shape the physical structures of the system to reflect and respond to information about the world that is relevant to it. The meaning of higher-order states is thus grounded in historical interaction with the environment and attuned to given selection criteria (e.g., in natural selection, the criteria is persistence and reproduction). We contend that a system whose activity is informed by this kind of higher-order meaning just *is* a system that is acting for reasons (*contra* vertical reductionism). Additionally, these reasons are rightfully understood as inhering at the level of the whole system because that is where the locus of fitness is that selects for them, that level is what those reasons are about, and so that is where the appropriate locus of meaning and causality lies (*contra* event causation).

This set of conditions, if collectively met, avoid all three of the arguments set out in the introduction—external determinism, vertical (causal) reductionism, and horizontal reductionism (including event-based causation). Moreover, it does so in a perfectly naturalistic way, without the supposed mysticism or dualism that accounts of agent causation have often been accused of. Perhaps most notably, it meets the challenge of Kim by showing that the causation in the system is *not* wholly inherent in or captured by the low-level details of all the physical components at any given moment. As living organisms demonstrate, any such picture of the instantaneous state of an agential system misses the extended history of causal influences that imbue the states with meaning relative to the goals of the system as a whole, meaning on which the agent selects its actions. Thus, it would be wrong to reduce causation to the system’s lowest level of description.

In setting out these criteria for the justification of agent causation *in principle*, we simultaneously argued that biological organisms, *in practice*, may satisfy most, if not all, of them. To reiterate, while we used living systems as model examples to help conceptualise how each condition could conceivably be met, we did not intend to imply that living organisms are necessarily the only things that could qualify as agents. The primary purpose of the paper was to set out a plausible way in which agent causation could be realised or naturalised in *any* theoretical system and in doing so, lay the groundwork for future research that uses the framework to evaluate the agency of individual organisms or systems.

Indeed, we take the criteria presented here to be dimensional, as opposed to categorical. We would therefore expect that agents vary considerably in how they satisfy each condition, with different systems perhaps performing strongly on some criteria but not on others. In this sense, existing mathematical methods and formalisms for measuring agency, and other related parameters such as autonomy, integration and consciousness, are well-suited to quantifying the sort of multidimensional agency we have outlined here. Different information-theoretic formalisms of *causal emergence*, for example, may be applied to measuring higher-order causation within a system [82,83,84,85]. Similarly, attempts to formalise notions of individuality [81,86] and autonomy [87,88] can be used to measure *how* endogenously active and free from determinate external forces a system is, by quantifying the degree to which it may be more “interested in itself rather than the world outside” [89] (p. 3). Finally, the criteria of holistic integration and informational causation more generally fits neatly with the long-standing measures associated with *Integrated Information Theory* (IIT) [90,91,92]. In sum, then, the framework presented here should help researchers working on measuring agency to situate their findings within the context of agent causation. If the analysis offered above is accurate, then agency is a multidimensional, multiply-realisable concept that cannot be quantified in terms of a single parameter such as thermodynamic autonomy, system-wide integration or emergent information. It is a composite concept, with systems likely exhibiting different ‘agency profiles’ to one another and possibly even differences across an individual’s lifespan.

## 4. Addendum: The Reductive Instinct

The idea that organisms can be causal agents, that they can act in the world, is entirely in keeping with common thinking. Even very young children naturally attribute agency to objects that appear to be moving under their own power and acting in an intentional manner [93,94]. The framework we outline above provides a naturalistic way to think about how such causal agency could emerge over phylogeny and ontogeny. Moreover, this view of agents, adopting the ‘intentional stance’, is invaluable and arguably indispensable in building explanatory theories of organismal behaviour [4,95,96]. Why then is this view seen as problematic in philosophical circles, and why do so many scientists tend to fall into a more reductive, mechanistic way of thinking that seems to eliminate agents from the picture?

Part of the reason may be a slippage from methodological into theoretical reductionism. Biologists investigating the processes of life naturally try to isolate single processes from the ongoing dynamics of the cell or organism and, further, try to isolate individual components of those processes to understand their functional roles and, ultimately, the logic of the entire process. To this end, they employ experimental techniques that powerfully manipulate individual components and measure some specific outcomes while attempting to hold as much of the background activity of the system constant. This horizontally reductive approach is (apparently at least) extremely powerful, for example, in delineating biochemical pathways or neural circuits mediating diverse cellular or organismal functions and in assigning roles to the many components of such subsystems.

This approach naturally lends itself to thinking that an organism’s components truly act in isolation from each other. Even in philosophy, there is a tendency to build logical propositions, normative frameworks, or tightly constrained thought experiments that consider properties or events in isolation when trying to understand the causal logic of a system or tease out individual causal determinants in absolutist terms [97].

However, just because it is possible and often useful to experimentally or conceptually isolate components or pathways or processes or properties, and to consider them separately, while holding everything else constant, does not mean that these elements actually “work” separately or have truly isolatable causal efficacy in the normal course of things. Even the use of a term such as “working” may give an overly mechanistic framing [98]. Reductive approaches foster an illusion of linear pathways with dedicated components. However, any such picture relies on a forced perspective. Adopting different experimental or conceptual perspectives usually reveals that the cellular components or neural circuits are functionally involved in many different functions, that they have more promiscuous interactions than revealed from a single angle, and that both their activity and the consequences of their activity are highly context-dependent, integrated with the activity of other components and subsystems.

The challenges in translating basic research findings obtained using these kinds of reductive approaches to the clinic [99], in areas from cancer to psychiatry to neurodegenerative disorders, highlight the degree of hubris in thinking that complex dynamical systems can truly be decomposed and that manipulations performed under controlled conditions in the lab will have equally predictable and controllable outcomes “in the wild”. The apparent successes of the reductive methodological approach thus need not, and we argue, should not entail a commitment to theoretical reductionism.

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
