# Peer review of "Naturalising Agent Causation"

_entropy, 2022, doi:10.3390/e24040472_

Round 1

Reviewer 1 Report

In this work, Potter and Mitchell provide a timely and concise account on behalf of causal agency as an evolved trait. Given the persistence with which reductionistic interpretations and engineering metaphors continue to dominate the literature, this work is clearly a 'heavy-hitter' with regards to rooting out all too common misperceptions and inconsistencies in the field. Given that my views and perspectives align very much with the authors', perhaps one ought to not place too much wieght on my assessment that this manuscript ought to be required reading for anybody interested in behavior, neuroscience or the philosophy of free will.

As this manuscript does not describe experimental work, it arguably does not require stringent peer-review. In such cases, the reviewers' tasks are to make sure the arguments are coherent, the relevant literature cited and other basic rules of scholarship are followed. The literature in this field is very large and so one can always argue about whether one or the other references ought to be added or left out. I have only very few quibbles wrt the literature, mentioned below. I even think another one of my own articles could have been referenced, but then again, given the size of the literature, there likely are hundreds of other articles the authors could have referenced as somehow related. Thus, by and large, the authors are citing the literature relevant to their line of reasoning and they do this very much according to current practice. I can also not find any other objective reasons why this article should not be published.

Therefore, I have to restrict my comments to suggestions and reader feedback, with the intent to improve the manuscript. Please find this feedback below.

I'll start with the end of the article: Their 'addendum' on the misleading consequences of 'practical reductionism' ought ot receive a more prominent caption, emphasizing that studying the components of a system is always a necessary step towards understanding it, but it surely is never sufficient, nor ought it be a goal in itself. In times of -omics it may be important to stress these points more than one perhaps otherwsie would?

Line 549: "In simple systems, the meaning of a signal can be pragmatically embodied through a direct coupling to some historically adaptive action. As organisms increased in complexity, especially with the evolution of multicellularity and nervous systems, perception and action became decoupled"

This evolutionary path is not quite as straightforward as it may seem or as the auhtors seem t imply. Citing Jékély et al.

https://royalsocietypublishing.org/doi/full/10.1098/rstb.2019.0764

one can just as well argue that the stimulus-response organization is actually a derived evolutionary trait and not ancestral to a re-afferent organization. I suggest the authors cite this more modern work instead of the somewhat dated references 64/65 and adjust the text in this paragraph accordingly.

Looking at the following paragraphs and to some extent also before (see below for a short list of possible examples), for an article that purports to develop a concept of agency, the reactive properties of organisms are surprisingly often used as examples in the arguments of the authors. Surely, more active examples of, e.g., problem-solving, trying out and exploration would be more fitting to describe the active properties of an agent?

What I am missing is more emphasis that in most cases, the agent does not receive sufficient input from the environment to be able to unambiguously determine the best next course of action, and yet, the metaphors of receiving input and then acting accordingly is pervasive in this work - despite the authors arguing for agency, rather than reactivity.

It seems to me that this repeated reference to the reactivity of agents (almost an oxymoron!) detracts from the arguments they are trying to make and unnecessarily weakens their position by evoking a passive-reactive image in the mind of their readers, when the authors are explicitly trying to evoke something that would appear more like the opposite. It appears to me that constantly referring to stimulus-response examples would appear to undercut the authors' own arguments?

For the same reasons, I would also recommend the authors consider carefully whether they describe such responses to external stimuli as "actions", when probably a more fitting term would be "reactions" (as the term "response" properly implies). Careful use of these terms may support the authors' goal of convincing readers to see the matter from their perspective. I've rcently been made aware that native English speakers may not necessarily understand the terms 'action' and 'reaction' as antonyms, but they are in other languages and the authors' goals may be better served if they also treated them as such, as much as the English language allows.

(I would like to add a more detailed reason for making a very conscious and explicit distinction between actions and reactions. According to current data, there may be very little difference neurobiologically between actions and reactions - they all seem to be actions, some with more, others with less input from the sensory areas from the nervous system leading up to their execution. Stimuli only very rarely, if at all, are both necessary and sufficient for any behavior to occur. The authors themselves emphasize this point. Once a behavior occurs, the general process by which it is executed may not differ very much whether there was a stimulus present or not. In fact, due to multiple realizeability, instance to instance variation in execution may well be larger than stimulus-nonstimulus variation. If one does not properly distinguish between actions and responses, one risks simply perpetuating the wrong concept of a response. Conversely, sharp distinction between actions and reactions leaves the outcome open: if one day actual responses were discovered, there would be a neurobiological class that deserves the moniker. If, on the other hand, current evidence were to prevail, one can state clearly that responses do not refer to a neurobiologically definable class of behaviors. Muddying the water between actions and reactions thus risks perpetuating old ideas, ideas against which this manuscript is directed.)

Some instances of potentially counter-productive stimulus-response examples:

Bacterial chemotaxis (l295-326 and 542) can be described in terms of orienting responses, but each tumble of the bacterium is of course an action: there is no particular stimulus to which the cell would 'respond" in a directed fashion. The authors write themselves (319): "the directional chemotactic response itself relies on temporal integration so that the bacterium responds not to absolute levels but to concentration changes as it move" What is happening is that the probability to tumble randomly increases as conditions deteriorate. These tumbles are neither elicited by any specific stimulus or concentration, nor are they directed towards or away from any specific stimulus. However, some of the wordings in these two paragraphs may confuse the reader not steeped in this process into assuming they are, as the descriptions of the authors here do not appear completely cionsistent.

Neurons are also often depicted as passive network components responding to input in some ways, e.g., lines 452, 469, 525 or 558. While some of these examples certainly are usefulfor the point the authors are trying to make, by emphasizing this passive-reactive property over the instances where neurons are, in fact, active drivers of neural activity, the authors may be unwillingly promoting passive-reactive concepts in the minds of their readers.

Perhaps most clearly counter-productive is the prevalence of passive-responsive examples in lines 573-5 ("historicity"), as if the accumulation of experiences only had an effect on how agents respond to stimuli and not on how they, e.g., change the way in which they try out solutions to new problems, or decide to explore new resources. In this respect, one may also perhaps consider concepts such as antifragility or allostasis as a consequence of the historicity of an organism, rather than how it changes the passive responses to changes in the environment.

Analogously, lines 636-38 describe responses to stimuli as the only/main property that nervous systems endow their organisms with, but then the paragraph contains references to actions as if actions were always responses to stimuli? Surely, the public discourse since Dickinson A (1985) Actions and Habits-the Development of Behavioral Autonomy. Philos Trans R Soc Lond B Biol Sci 308: 67–78. has progressed significantly beyond equating actions and responses, I hope?

I will refrain from listing any further examples now. I'm not saying the auhtors must replace all of their examples. What I am saying is that perhaps the authors may want to reconsider if emphasizing passive-reactive organizations and examples really is the best way to convince readers that active-dynamic agents do exist and that they are a product of evolution that can be studied empirically.

Author Response

Response to Reviewer 1: Potter and Mitchell 9th March 2022

We would like to thank the reviewer for their positive response to our paper “Naturalising Agent Causation”, and for their incisive and helpful suggestions. We appreciate the time and thought that went into the review and believe that, as a consequence, the revised manuscript now represents a much clearer and more precise version of the article. We have incorporated most of the suggestions made in the review. Those changes are highlighted within the manuscript.

The main criticism, as we understood it, was our choice of examples and our lack of precision with the use of the word “response”, as compared with “action”. These points were well made and we particularly appreciated the insight into how these terms may be perceived by non-native English speakers. We have thus made a number of changes to the manuscript in order to be more careful with the language we are using and thereby, hopefully, avoid undermining our overall argument in the ways highlighted by the review.   

In particular, we have re-written large parts of the bacterial chemotaxis (lines 336-368) and neuronal communication (lines 582-595) sections in order to emphasise the active and endogenous nature of these systems and behaviours. On this point, we’d like to clarify that our rationale for choosing these examples is exactly the one described in the review; namely, to make the argument that biological behaviours are almost never “passive-reactive”, but almost always active and internally driven. In our opinion, taking examples from some of nature’s ‘simplest’ behaviours – which are commonly described and understood as being ‘passive-reactive’, stimulus-response behaviours – and demonstrating that they are, in fact, not that way at all is a strong way to make this argument. 

That said, we are grateful to reviewer 1 for helping us to appreciate that, by using the terms “response” and “reaction” when describing the actions in our examples, we may have been confusing the issue by appearing to continue with a passive-reactive interpretation of these behaviours, even though we are actually arguing for the exact opposite. We feel the changes we have made to the manuscript clarify our position on this, without needing to change the specific examples being used.

We believe the changes we have made in response to these insightful comments have greatly helped strengthen and clarify the manuscript. Please also see below, in blue, for a point-by-point reply to the review’s other comments and concerns. (All page numbers refer to the revised manuscript file with tracked changes.)

Comment 1: "In simple systems, the meaning of a signal can be pragmatically embodied through a direct coupling to some historically adaptive action. As organisms increased in complexity, especially with the evolution of multicellularity and nervous systems, perception and action became decoupled". This evolutionary path is not quite as straightforward as it may seem or as the authors seem to imply. Citing Jekely et al., one can just as well argue that the stimulus-response organization is actually a derived evolutionary trait and not ancestral to a re-afferent organization. I suggest the authors cite this more modern work instead of the somewhat dated references 64/65 and adjust the text in this paragraph accordingly.”

Author Response: Thank you for this point. As suggested, we have included the Jékely et al., (2021) reference, as well as related work by Keijzer et al., (2013) and Dupre & Yuste (2017). We have also adjusted the text in that paragraph to emphasise that “the first nervous systems…may have co-evolved with muscles to allow coordinated movement of the various parts of multicellular creatures. Linking these to sensory receptors could then allow appropriate responses to be selected based on information about the external world, in the context of the animal’s own movements”. (lines 715-720)

Comment 2: “What I am missing is more emphasis that in most cases, the agent does not receive sufficient input from the environment to be able to unambiguously determine the best next course of action.”

Author Response: Thank you for this point. We agree that this is a useful angle for the argument against external determinism, which we were previously missing. We added in a paragraph making the point that “often, external inputs are simply not rich enough for external determinism to hold” and so instead organisms must take epistemic foraging actions to probe the environment and try out new problem-solving solutions (lines 295-302).

Comment 3: Our discussion of Historicity seemed to suggest that “the accumulation of experiences only [has] an effect on how agents respond to stimuli and not on how they, e.g., change the way in which they try out solutions to new problems, or decide to explore new resources…. Analogously, lines 636-38 describe responses to stimuli as the only/main property that nervous systems endow their organisms with”

Author Response: Thanks, again, for pointing this out. You’re right that the way in which the section was framed may have implied that natural selection and learning only shape the way in which an organism behaves in the presence of a specific stimulus. To clarify this, we added in a paragraph explaining that historicity can also endow agents with more abstract behaviours (e.g. “learning what to learn from, what information to seek out, and what active tactics and strategies to use to best pursue their goals”) (line 780-784).

Reviewer 2 Report

Please see attached for my review.

Author Response

Response to Reviewer 2: Potter and Mitchell 9th March 2022

We would like to thank reviewer 2 for their very positive comments on our paper “Naturalising Agent Causation”, and their helpful literature suggestions. We appreciate the time and thought that went into the review and believe that, as a consequence, the revised manuscript is now a far richer reflection of the surrounding literature. We have incorporated most of the suggestions made in the review. Those changes are highlighted within the manuscript and summarised below. (All page numbers refer to the revised manuscript file with tracked changes.) 

  • We added an explicit mention of Peter Tse’s work on Criterial Causation (lines 584-585) and included multiple citations to his book The Neural Basis Of Free Will.
  • We included reference to Sinnott-Armstrong’s 2019 Contrastive Mental Causation article with regard to the concept of “sensitivity”. (lines 552-554)
  • We included two references to the Free Energy/Active Inference/Predictive Processing literature, via Andy Clark’s Surfing Uncertainty book and Giovanni Pezzulo’s article on epistemic foraging and active inference (lines 301-302). These were added in the context of predictive inference and active perception, however, and not explicitly “boundaries and self-making” as the review had initially suggested. We chose not to mention the Free Energy Principle and active inference framework in these contexts because of the current scrutiny and confusion (at least on our part) surrounding what the FEP has to say about system boundaries (e.g. see Bruineberg et al., (2021) The Emperor’s New Markov Blanket, BBS article and commentaries).
  • We briefly discussed Joslyn’s (1999) point that semiotic processes in complex systems are contingent and arbitrary (lines 731-735).
  • We referenced Dohmatob et al., (2020) in support of the view that the brain is endogenously active. (lines 288)
  • We chose not to include the suggested literature on Compatibilism because the intention of this paper is to focus on agency in general and not free will in humans. We have added a footnote on page 4 highlighting this point and clarifying what we take the relevant distinction to be.

Thanks again to the reviewer for these suggestions. We feel these changes have strengthened the manuscript and helped to better situate our research within the wider literature.

Round 2

Reviewer 1 Report

The authors have adequately addressed the points I raised.

Björn Brembs

Reviewer 2 Report

This already excellent (and important) manuscript has been made even better.

I believe is ready to be published as is, but I will provide a few additional comments in case the authors would like to make any additional revisions:

While FEP-AI is still an evolving paradigm, and while many of the concepts involved have an older vintage, I believe the authors will be curious to know that new analytic techniques are being developed that will be better able to handle multi-scale dynamic boundaries and speak to present critiques. Still, the authors may be interested in this work:

The Markov blankets of life: autonomy, active inference and the free energy principle | Journal of The Royal Society Interface (royalsocietypublishing.org)

I also have some lingering concerns about emphasizing indeterminacy on the level of 'fundamental' physics, as I believe the arguments made by the authors have made it such that a case for agency/autonomy has been made that could be "compatible" with deterministic interpretations of quantum mechanics. While this issue doesn't need to be resolved here, considering the importance of the topic, I would feel some relief if the authors included a statement indicating that these alternative perspectives exist in physics, and that their arguments would hold even if something like hidden-variable interpretations were found to be the best reflection of physical reality.

I thought the discussion linking normativity to the agent-level via natural selection (in terms of phylogeny, rather than generalized evolution, which perhaps could be said to 'explain' almost everything) was profound. I wonder if it could be helpful to add an additional sentence to emphasize learning --> agent-level concerns around here: "On both evolutionary and individual timescales, the agent is thus the locus of fitness. It thereby becomes both the locus of meaning and also the locus of control that is informed by and evaluated relative to that meaning." I'd also be interested in a brief footnote regarding units of selection debates (e.g. multi-level evolution and conditions under which group selection might be significant), although perhaps that might distract from the main message.

Finally, at some point I'd be grateful if the authors could provide feedback to the author of this manuscript, who has a (I believe value-aligned) interest in similar issues, and is a strong supporter of this work:

Entropy | Free Full-Text | The Radically Embodied Conscious Cybernetic Bayesian Brain: From Free Energy to Free Will and Back Again | HTML (mdpi.com)